# Retrieval-based Knowledge Transfer: An Effective Approach for Extreme Large Language Model Compression

**Jiduan Liu[1,2*], Jiahao Liu[3*], Qifan Wang[4] , Jingang Wang[3], Xunliang Cai[3]**
**Dongyan Zhao[1,2,5,6†], Ran Lucien Wang[7] , Rui Yan[7†]**

[1]Wangxuan Institute of Computer Technology, Peking University
[2]Center for Data Science, AAIS, Peking University; [3]Meituan; [4]Meta AI
[5]National Key Laboratory of General Artificial Intelligence; [6]BIGAI, Beijing, China
[7]Gaoling School of Artificial Intelligence, Renmin University of China
{liujiduan,zhaody}@pku.edu.cn, ruiyan@ruc.edu.cn, wqfcr@fb.com
{liujiahao12,wangjingang02,caixunliang}@meituan.com, ran.wang.math@gmail.com

## Abstract

Large-scale pre-trained language models (LLMs) have demonstrated exceptional performance in various natural language processing (NLP) tasks. However, the massive size of these models poses huge challenges for their deployment in real-world applications. While numerous model compression techniques have been proposed, most of them are not well-suited for achieving extreme model compression when there is a significant gap in model scale. In this paper, we introduce a novel compression paradigm called Retrieval-based Knowledge Transfer (RetriKT), which effectively transfers the knowledge of LLMs to extremely small-scale models (e.g., 1%). In particular, our approach extracts knowledge from LLMs to construct a knowledge store, from which the small-scale model can retrieve relevant information and leverage it for effective inference. To improve the quality of the model, soft prompt tuning and Proximal Policy Optimization (PPO) reinforcement learning techniques are employed. Extensive experiments are conducted on low-resource tasks from SuperGLUE and GLUE benchmarks. The results demonstrate that the proposed approach significantly enhances the performance of small-scale models by leveraging the knowledge from LLMs.

## 1 Introduction

Pre-trained language models (PLMs), such as BERT/RoBERTa (Devlin et al., 2019; Liu et al., 2019), have demonstrated exceptional performance across various natural language processing (NLP) applications. However, these models typically comprise hundreds of millions of parameters, presenting a substantial challenge for researchers due to their massive scale. As a result, the full potential of large-scale pre-trained language models (PLMs) remains untapped. To tackle this challenge, a multitude of model compression techniques have been proposed, encompassing knowledge distillation (Sanh et al., 2019; Jiao et al., 2020; Passban et al., 2021), network pruning (Liang et al., 2021a; Gordon et al., 2020), quantization (Zhang et al., 2020; Tao et al., 2022) and weight sharing (Lan et al., 2020).

However, these model compression methods are not directly applicable to scenarios requiring high compression ratios, such as knowledge distillation. In such cases, the introduction of assistant models (Mirzadeh et al., 2020; Son et al., 2021) often leads to decreased and unstable performance. Recently, there has been a growing interest in leveraging large language models (LLMs) (Touvron et al., 2023; Zeng et al., 2022; Ouyang et al., 2022; Scao et al., 2022) that possess extensive language knowledge and can be effectively employed in various downstream tasks. Consequently, it is essential to explore methods for transferring this knowledge to small-scale models. Nonetheless, existing approaches are inadequate for compressing LLMs due to their exceptionally high compression ratios. Some prior research (Wang et al., 2022; Dai et al., 2023; Ubani et al., 2023) has suggested utilizing LLMs for data augmentation and knowledge transfer to small-scale models, which allows the latter to demonstrate improved performance on low-resource datasets. However, when tackling more challenging tasks like the SuperGLUE benchmark (Wang et al., 2019a), the limited parameter size of small-scale models becomes a hindrance, preventing them from effectively retaining the knowledge transferred by LLMs. Consequently, the performance enhancement achieved for small-scale models remains constrained.

---

[*]Equal contribution.
[†]Corresponding authors: Dongyan Zhao (zhaody@pku.edu.cn) and Rui Yan (ruiyan@ruc.edu.cn).

To effectively transfer the knowledge of Large Language Models (LLMs) to small-scale models, enabling them to efficiently and accurately complete tasks, we propose a novel compression paradigm called Retrieval-based Knowledge Transfer (RetriKT). Our approach involves two main steps: knowledge extraction from the LLM to construct a knowledge store, and subsequent retrieval of relevant information from the knowledge store by the small-scale model to accomplish the task. More specifically, we employ the technique of soft prompt tuning to fine-tune an LLM, ensuring that it generates in-domain samples. Additionally, we introduce the Proximal Policy Optimization (PPO) (Schulman et al., 2017) reinforcement learning algorithm to enhance the generation quality. As a final step, the small-scale model learns to retrieve pertinent information from the knowledge store. We perform an extensive set of experiments on truly low-resource and challenging tasks sourced from SuperGLUE (Wang et al., 2019a) and GLUE (Wang et al., 2019b) benchmarks. The experimental results demonstrate that RetriKT significantly enhances the performance of small-scale models and outperforms previous SOTA knowledge distillation methods by leveraging the knowledge of LLMs. This indicates the effectiveness and practicality of the retrieval-based knowledge transfer paradigm for extreme model compression.

Our contributions can be summarized as follows:

- We propose a new compression paradigm called Retrieval-based Knowledge Transfer, which aims to transfer knowledge from LLMs to extremely small-scale models. This paradigm addresses the challenge of achieving extreme model compression when there is a significant gap in model scale.

- We introduce the reinforcement learning algorithm PPO to enhance the generation quality, and carefully design the reward function. This technique contributes to improving the diversity and accuracy of the knowledge extracted from LLMs used for knowledge transfer.

- We conduct extensive experiments on low-resource tasks from the SuperGLUE and GLUE benchmarks. The results demonstrate that RetriKT significantly enhances the performance of small-scale models and outperforms previous SOTA knowledge distillation methods by leveraging the knowledge from LLMs.

## 2 Related Work

This paper involves three areas of work: knowledge distillation, reinforcement learning, and data augmentation.

### 2.1 Knowledge Distillation

We first introduce related work of knowledge distillation (KD), which can be categorized as response-based, feature-based, and relation-based KD. Response-based KD was initially introduced by Hinton et al. (2015), which transfers label knowledge by minimizing the KL-divergence between predicted distributions of the teacher and the student. Building upon this concept, Sanh et al. (2019); Liang et al. (2021b) applied response-based KD to tasks such as masked language modeling and text classification, resulting in smaller models with slight performance drops. Feature-based approaches which are initially introduced by Romero et al. (2015), entail aligning the feature activations between the teacher and the student models. Building upon this concept, more recent techniques have expanded the scope by incorporating hidden representations of the [CLS] token as indicators (Sun et al., 2019), matching hidden representations of all tokens (Jiao et al., 2020), and introducing customized feature-based distillation (Sun et al., 2020). On the other hand, relation-based methods (Park et al., 2021; Liu et al., 2022a; Jiao et al., 2020; Wang et al., 2020, 2021a; Wu et al., 2023) aim to emphasize the importance of capturing and utilizing relations among multi-granularity representations in both horizontal and vertical directions.

### 2.2 Reinforcement Learning

Recently, reinforcement learning (RL) has gained significant attention in the field of language modeling. This approach has been successfully applied to various language tasks, including summarization (Paulus et al., 2018; Ziegler et al., 2019; Stiennon et al., 2020; Wu et al., 2021), dialogue systems (Zhou et al., 2017; Jaques et al., 2020; Hancock et al., 2019), machine translation (Bahdanau et al., 2017; Kreutzer et al., 2018; Kiegeland and Kreutzer, 2021), semantic parsing (Lawrence and Riezler, 2018), story generation (Zhou and Xu, 2020), and question generation (Pang and He, 2021). Concurrently, there has been a growing interest in using RL to align language models with human preferences across a wide range of language tasks. For example, Ouyang et al. (2022) employed

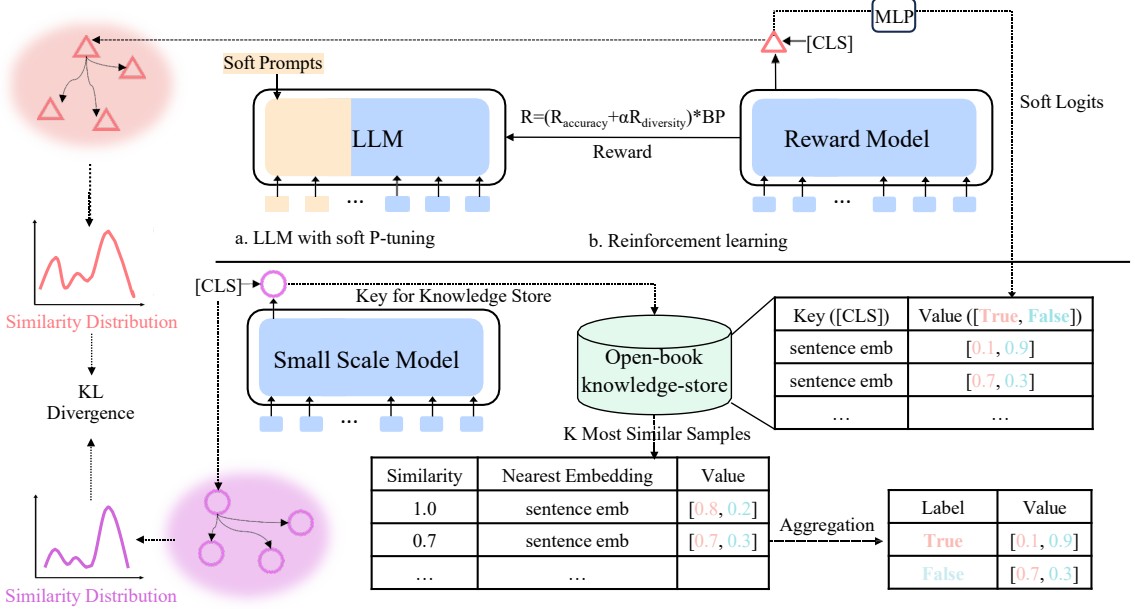

Figure 1: The framework of RetriKT which consists of three steps: (1) fine-tune additional soft prompts for the LLM by supervised learning; (2) further fine-tune the soft prompts by reinforcement learning to enhance the generation quality; (3) create the knowledge store based on the knowledge extracted from the LLM, and teach the small-scale model how to retrieve relevant knowledge from it. (Retrieval-based Knowledge Transfer)

RL techniques, specifically Proximal Policy Optimization (PPO) (Schulman et al., 2017), to fine-tune a large language model and align it with models of human preference.

## 2.3 Data Augmentation

Wu et al. (2019) and Kumar et al. (2020) have introduced a method to create synthetic data by randomly masking words in the original training instances. Another works (Ding et al., 2020; Yang et al., 2020; Anaby-Tavor et al., 2020) involved utilizing Language Models (LMs) and Pre-trained Language Models (PLMs) to directly generate synthetic data for NLU tasks. Wang et al. (2021b, 2022) proposed the use of hard prompts and soft prompts to generate synthetic data. In a related study, Liu et al. (2021b); Zan et al. (2022) have uncovered the complementary nature of PLMs and classical approaches.

## 3 Methodology

Our approach revolves around extracting knowledge from the parameters of a Large Language Model (LLM) and leveraging it for the benefit of an extreme small-scale model through retrieval. We introduce the technique to extract in-domain knowledge from the LLM for each task by freezing its parameters and fine-tuning additional soft

prompts using the corresponding dataset (section 3.1). To further enhance the generation quality, we introduce reinforcement learning and carefully design the reward function to fine-tune the soft prompts (section 3.2). Subsequently, we extract the in-domain knowledge from the LLM and create a knowledge store (section 3.3). Finally, we enable the small-scale model to retrieve and effectively utilize the knowledge generated by the LLM to successfully perform specific tasks (section 3.4).

## 3.1 Soft Prompt Tuning

To enhance the extraction of in-domain knowledge from the LLM, it is crucial to introduce appropriate prompts that precede the inputs, effectively guiding the text generation process (e.g. "Generate rte examples according to the following keywords:"). However, relying solely on manual design often leads to limited diversity and effectiveness in the generated knowledge. To overcome such limitations, we employ trainable parameters known as soft prompts, thereby replacing manual templates. We only need to update these soft prompts and keep all other parameters of the LLM fixed during training for each specific task.

More specifically, we utilize P-tuning v2 (Liu et al., 2022b), which involves incorporating a soft prompt into each Transformer (Vaswani et al.,

2017) layer. The soft prompt is represented as a sequence of trainable vectors, denoted as $P^j = \{p_1^j, \ldots, p_k^j\}$, where $j$ is the $j$-th layer and $k$ is a hyperparameter which represents the prompt length. The $i$-th hidden states at the $j$-th layer, denoted as $h_i^j$, can be expressed as follows:

$$
h_i^j = \begin{cases} p_i^j & i \leq k \\ w_i & i > k \wedge j = 0 \\ Trans(H^{j-1})_i & \text{Else} \end{cases} \quad (1)
$$

where $Trans()$ represents the Transformer layer, $H^j$ represents all hidden states at $j$-th layer and $w_i$ is the word embedding of the input text.

In order to generate diverse samples, we adopt the approach presented in PromDA (Wang et al., 2022), which involves generating samples from both the *Input View* and the *Output View*. The *Input View* is conditioned on the keywords present in the input texts, which are extracted by the unsupervised keyword extraction algorithm Rake (Rose et al., 2010). On the other hand, the *Output View* is conditioned on the labels. Additionally, we train two sets of soft prompts, namely $P_{input}$ and $P_{output}$, which are specifically designed to generate samples from the *Input View* and the *Output View* respectively. Each sample $S = (Y, X)$ generated by the LLM consists of two parts, namely the label $Y = \{y_i\}_{i=1}^{l_y}$ and the input text $X = \{x_i\}_{i=1}^{l_x}$. The supervised learning object for soft prompts $P_{input}$ and $P_{output}$ can be formalized as:

$$
\mathcal{L}_{\text{input}} = -\sum_{i=1}^{l_y + l_x} \log(p(s_i|s_{<i}, P_{input}, K)), \quad (2)
$$

$$
\mathcal{L}_{\text{output}} = -\sum_{i=1}^{l_y + l_x} \log(p(s_i|s_{<i}, P_{output}, Y)), \quad (3)
$$

where $K$ and $Y$ represent the keywords of the input text and the label, respectively.

## 3.2 Reinforcement Learning

Constructing a knowledge store that combines accuracy and diversity is essential for enabling the small-scale model to effectively retrieve relevant information. However, traditional supervised learning methods are unable to optimize these learning objectives due to the lack of per-token differentiability. Therefore, we employ reinforcement learning to improve the generation quality. This approach involves guaranteeing that the generated texts align with the labels, thereby ensuring the

accuracy of the knowledge store. Additionally, it ensures that the generated samples exhibit diversity, thus fulfilling the diversity requirement of the knowledge store.

More specifically, we utilize the original dataset to train a classification model, which serves as the reward model denoted as $RM()$. For each generated sample, comprising both input text and label, we assess the confidence score of the text's label using the reward model, thereby obtaining the accuracy reward $R_{\text{accuracy}}$. Additionally, we evaluate the diversity of the generated samples by employing the Self-Bleu metric (Zhu et al., 2018), which calculates the Bleu score (Papineni et al., 2002) considering other samples as references. Subsequently, we calculate the diversity reward $R_{\text{diversity}}$ as $1 - b_3$, where $b_3$ represents the Bleu-3 metric. To avoid the generation of overly simplistic patterns by the LLM, we apply a length penalty. If the length of a generated sample falls below a certain threshold $l_{\min}$, the reward is multiplied by a discount factor:

$$
\text{BP} = \begin{cases} 1 & l \geq l_{\min} \\ \exp(1 - \frac{l_{\min}}{l}) & l < l_{\min} \end{cases} \quad (4)
$$

where $l$ represents the length of the generated sample. The final reward for each generated sample is:

$$
R = (R_{\text{accuracy}} + \alpha R_{\text{diversity}}) \times \text{BP}, \quad (5)
$$

where $\alpha$ is a hyperparameter to balance the weight of accuracy reward and diversity reward.

Following previous studies (Ouyang et al., 2022; Bai et al., 2022), we employ Proximal Policy Optimization (PPO) (Schulman et al., 2017) as the reinforcement learning algorithm to fine-tune the soft prompts $P_{input}$ and $P_{output}$. To ensure reliable performance in generating in-domain samples, we incorporate supervised gradients into the PPO gradients. Our objective is to minimize the following loss function:

$$
\mathcal{L}_{\text{LLM}} = \mathcal{L}_p + \mathcal{L}_v + \beta * \mathcal{L}_{\text{sft}}, \quad (6)
$$

where $\mathcal{L}_p$ and $\mathcal{L}_v$ are actor loss and critic loss of the PPO algorithm respectively, $\mathcal{L}_{\text{sft}}$ represents $\mathcal{L}_{\text{input}}$ and $\mathcal{L}_{\text{output}}$ for $P_{input}$ and $P_{output}$ respectively, and $\beta$ is the hyperparameter which controls the strength of supervised gradients.

### 3.3 Knowledge Store

We employ the fine-tuned LLM to generate samples and construct a knowledge store for each task. Specifically, for each sample in the original dataset $D$, we utilize the keywords extracted by the Rake algorithm and labels as inputs for the model. Each sample is used $m$ times, and in each iteration, $n$ samples are generated. To ensure distinctiveness among the generated samples, we adopt p-sampling (Holtzman et al., 2020), which constrains tokens to the smallest possible set whose cumulative probability exceeds the probability parameter $p$. The set of samples generated from keywords is denoted as $D_\text{I}$, while the set generated from labels is denoted as $D_\text{O}$. The quantity of both $D_\text{I}$ and $D_\text{O}$ is $mn|D|$, where $|D|$ represents the quantity of the original dataset $D$.

To further enhance the diversity of generated samples, we utilize the label of each sample in $D_\text{I}$ as the model input to generate a sample each time, resulting in the sample set $D_\text{IO}$. Similarly, we use the keyword of each example in $D_\text{O}$ as the model input to generate a sample each time, resulting in the sample set $D_\text{OI}$. The final knowledge store is composed of these sample sets, namely $D \bigcup D_\text{I} \bigcup D_\text{O} \bigcup D_\text{IO} \bigcup D_\text{OI}$, which theoretically yields a total of $(1 + 4mn)|D|$ samples. However, there is a possibility of duplicates in the generated samples, so we remove them.

As illustrated in Figure 1, the key for each sample in the knowledge store is the sentence embedding produced by the small-scale model. These embeddings enable the model to retrieve relevant information from the knowledge store. The value of each sample is the probability distribution produced by the reward model $RM()$ for each task.

### 3.4 Retrieval-based Knowledge Transfer

Due to the limited number of parameters in the small-scale model, it is not feasible to directly memorize all the knowledge extracted from the LLM. Therefore, we do not train the small-scale model directly with the generated samples. Instead, we leverage the knowledge in a retrieval manner. Specifically, we utilize the reward model $RM()$ to provide the sentence representation $f^T(x_i)$ for each generated sample $x_i$, which is then used to calculate cosine similarity scores with other generated samples in the mini-batch. The similarity score list obtained from the reward model can be denoted as $S_i^T = \{s^T(x_i, x_j)\}_{j \in [1,N] \wedge j \neq i} =$

| Dataset | \|Train\| | \|Dev\| | \|Test\| | Metrics |
|---------|-------|------|-------|---------|
| BoolQ | 9427 | 3270 | 3245 | acc. |
| CB | 250 | 56 | 250 | acc. |
| COPA | 400 | 100 | 500 | acc. |
| RTE | 2490 | 277 | 3000 | acc. |
| WiC | 5428 | 638 | 1400 | acc. |
| Cola | 8551 | 1043 | 1063 | mcc. |

Table 1: The data statistics and metrics of all tasks.

$\{\phi(f^T(x_i), f^T(x_j))\}_{j \in [1,N] \wedge j \neq i}$, where $N$ represents the batch size and $\phi$ denotes the cosine similarity. Similarly, we can obtain the similarity score list $S_i^S$ from the small-scale model. These similarity scores are then normalized in a listwise way to obtain the relevance distributions:

$$\tilde{s}^T(x_i, x_j) = \frac{e^{s^T(x_i, x_j)/\tau_1}}{\sum_{k \in [1,N] \wedge k \neq i} e^{s^T(x_i, x_k)/\tau_1}}, \quad (7)$$

$$\tilde{s}^S(x_i, x_j) = \frac{e^{s^S(x_i, x_j)/\tau_2}}{\sum_{k \in [1,N] \wedge k \neq i} e^{s^S(x_i, x_k)/\tau_2}}, \quad (8)$$

where $\tau_1$ and $\tau_2$ are temperature hyperparameters to smooth the distributions.

Our objective is to enable the small-scale model to learn how to retrieve relevant information from the knowledge store. To achieve this, we minimize the KL-divergence between the two relevance distributions $\tilde{S}_i^T = \{\tilde{s}^T(x_i, x_j)\}_{j \in [1,N] \wedge j \neq i}$ and $\tilde{S}_i^S = \{\tilde{s}^S(x_i, x_j)\}_{j \in [1,N] \wedge j \neq i}$ as the learning object for the small-scale model:

$$\mathcal{L}_\text{small} = \sum_i^N \tilde{S}_i^S \log \frac{\tilde{S}_i^S}{\tilde{S}_i^T}. \quad (9)$$

During the inference of the small-scale model, we retrieve the k most relevant samples from the knowledge store for each test sample $x$, denoted as $\{x_i\}_{i=1}^k$. To determine the weight of each retrieved sample $x_i$, we calculate the similarity score between the test sample $x$ and the retrieved sample $x_i$. The final prediction logit score for each class of the small-scale model is obtained as follows:

$$p_c^S = \frac{\phi(x, x_i)}{\sum_{j=1}^k \phi(x, x_j)} p_c^T, \quad (10)$$

where $c$ is a specific class and $p_c^T$ is the confidence score of the label $c$ predicted by the reward model $RM()$. The final prediction of the small-scale model is the class with the largest logit score.

| Model | #Params | WiC | CB | COPA | RTE | BoolQ | CoLA | Avg. |
|---|---|---|---|---|---|---|---|---|
| EncT5$_{xl}$ (Teacher) | 3B | 75.71 | 98.20 | 92.00 | 92.06 | 88.99 | 71.63 | 86.43 |
| | | | | | | | | |
| BERT$_2$ (Student) | 7.2M | 59.72 | 78.57 | 55.00 | 61.37 | 68.81 | 23.35 | 57.80 |
| + Vanilla KD (Hinton et al., 2015) | 7.2M | 60.66 | 78.57 | 56.00 | 62.09 | 68.72 | 27.14 | 58.86 |
| + Vanilla KD (w. TA) (Hinton et al., 2015) | 7.2M | 61.13 | 82.14 | 58.00 | 61.37 | 69.33 | 26.12 | 59.68 |
| + MSGKD (w. TA) (Liu et al., 2022a) | 7.2M | 59.72 | 83.93 | 60.00 | 62.45 | 68.13 | 29.39 | 60.60 |
| + AD-KD (w. TA) (Wu et al., 2023) | 7.2M | 62.85 | 83.93 | 61.00 | 62.82 | 69.60 | 31.89 | 62.02 |
| + RetriKT-KD | 7.2M | 63.79 | 89.29 | 63.00 | 64.26 | 71.10 | 40.94 | 65.40 |
| + RetriKT-Retrieval | 7.2M | **66.61** | **94.64** | **66.00** | **66.43** | **74.62** | **47.87** | **69.36** |
| | | | | | | | | |
| BERT$_4$ (Student) | 13.5M | 62.85 | 85.71 | 60.00 | 65.34 | 70.86 | 38.69 | 63.91 |
| + Vanilla KD (Hinton et al., 2015) | 13.5M | 62.70 | 85.71 | 62.00 | 66.43 | 70.73 | 39.16 | 64.46 |
| + Vanilla KD (w. TA) (Hinton et al., 2015) | 13.5M | 63.79 | 87.50 | 60.00 | 66.06 | 70.98 | 40.81 | 64.86 |
| + MSGKD (w. TA) (Liu et al., 2022a) | 13.5M | 62.38 | 87.50 | 63.00 | 65.34 | 71.13 | 40.78 | 65.02 |
| + AD-KD (w. TA) (Wu et al., 2023) | 13.5M | 64.89 | 87.50 | 64.00 | 66.43 | 72.81 | 41.77 | 66.23 |
| + RetriKT-KD | 13.5M | 68.65 | 92.86 | 72.00 | 74.01 | 77.22 | 58.16 | 73.82 |
| + RetriKT-Retrieval | 13.5M | **69.44** | **94.64** | **74.00** | **75.45** | **78.75** | **60.77** | **75.51** |

Table 2: Main experimental results (%) on six low-resource tasks. We also report the quantity of parameters of each PLM (without embeddings). We re-implement all baseline models based on the released code provided by AD-KD (Wu et al., 2023), and incorporate BERT$_{base}$ as the TA model (w. TA). There are two versions of RetriKT, one of which is trained by Vanilla KD by generated samples (RetriKT-KD), and the other is trained by retrieval learning object (RetriKT-Retrieval). The best results of each backbone are shown in bold. Results are statistically significant with respect to all baselines on each student model (all p-value < 0.005).

## 4 Experimental Setup

### 4.1 Datasets

We assess the performance of our method on multiple datasets from the SuperGLUE beachmark (Wang et al., 2019a) and GLUE benchmark (Wang et al., 2019b), including BoolQ (Clark et al., 2019), CB (De Marneffe et al., 2019), COPA (Roemmele et al., 2011), RTE, WiC (Pilehvar and Camacho-Collados, 2019), CoLA (Warstadt et al., 2019). These datasets pose a challenge for small-scale models as they are considered difficult and low-resource, with training samples numbering fewer than 10K (details in Table 1).

### 4.2 Baselines

We compare our methods with Vanilla KD (Hinton et al., 2015) and recent strong KD methods including MSGKD (Liu et al., 2022a) and AD-KD (Wu et al., 2023), which are re-implemented based on the released code provided by AD-KD. We fine-tune EncT5$_{xl}$ (Liu et al., 2021a) as the teacher model, and train BERT$_{base}$ using the Vanilla KD method as the assistant model (TA) for all baselines. We carry out grid-search of learning rate $\in$ {1e-5, 2e-5, 3e-5, 4e-5}, batch size $\in$ {4, 8, 16}

for datasets COPA and CB, and batch size $\in$ {16, 32, 64} for other datasets. The training epoch is set to 40 for CB and COPA, 20 for RTE and 10 for other datasets. We present the results on the validation set obtained from the best checkpoint during training.

### 4.3 Implementation

We implement all experiments with the deep learning framework PyTorch based on PromDA (Wang et al., 2022) and trl library (von Werra et al., 2020) on up to eight NVIDIA Tesla A100 GPUs (80GB memory). We build the LLM based on T5$_{xl}$ with 3B parameters (Raffel et al., 2020), and translate each dataset into a single-sentence format according to the template provided by T5. The pre-processed example of each dataset is shown in Appendix B. We utilize two small-scale BERT models released by Turc et al. (2019), one with 4 Transformer layers, 512 hidden neurons and 8 attention heads, and the other with 2 Transformer layers, 512 hidden neurons and 8 attention heads.

First, we tune the soft prompts $P_{input}$ and $P_{output}$ by supervised learning as the reference and initial models for PPO algorithm. The prompt length is set to 8 for all datasets. The learning

rate and batch size is set to 1e-3 and 64, respectively. The training step is set to 10K for COPA and CB, 20K for RTE, and 40K for other datasets. We fine-tune EncT5$_{xl}$ as the reward model for each dataset, which is also the teacher model for all baseline models for a fair comparison. Then we train the soft prompts through a combination of supervised learning and reinforcement learning by the loss function defined in Eq.(6). We set $\alpha$ and $\beta$ to 0.2 and 1 respectively. Generate number $n$ and probability $p$ for top-p sampling are always set to 5 and 0.9 respectively, while sample time $m$ is set to 64 for COPA and CB, 16 for RTE, and 8 for other datasets. Finally, we train the small-scale BERT models with a grid search of learning rate $\in$ {2e-5, 3e-5}, temperate combinations $(\tau_1, \tau_2) \in$ {(0.2, 0.1), (0.1, 0.05)}. The batch size is set to 128. There are two versions of RetriKT, one of which is trained by Vanilla KD by generated samples (RetriKT-KD), and the other is trained by retrieval learning object (RetriKT-Retrieval). More training details for PPO algorithm can be found in Appendix A

## 5 Experimental Results and Analysis

### 5.1 Main Results

As presented in Table 2, it is evident that both RetriKT-KD and RetriKT-Retrieval outperform previous KD methods across all backbones, which demonstrates the effectiveness of our approach. For instance, compared to the previous state-of-the-art method AD-KD, RetriKT-Retrieval demonstrates significant improvements: 7.34% on BERT$_2$ and 9.28% on BERT$_4$. Notably, RetriKT-Retrieval consistently outperforms RetriKT-KD, especially on BERT$_2$, indicating that retrieving relevant information from the knowledge store is more suitable for small-scale models rather than attempting to memorize all the knowledge. Furthermore, we observe comparable performance between Vanilla KD and Vanilla KD (w.TA), suggesting limitations in using the TA model for knowledge transfer. A more efficient approach for transferring knowledge from the LLM to the small-scale model is to prompt the LLM to generate in-domain knowledge, which can be retrieved by the small-scale model as relevant information for effective inference.

### 5.2 Ablation Study

To investigate the impact of different components of reinforcement learning in our approach, we con-

| PLM | Model | CB | COPA | CoLA |
|---|---|---|---|---|
| BERT$_2$ | RetriKT-KD | 89.29 | 63.00 | 40.94 |
| | w/o RL | 87.50 | 61.00 | 36.38 |
| | w/o R$_{accuracy}$ | 87.50 | 60.00 | 40.53 |
| | w/o R$_{diversity}$ | 85.71 | 62.00 | 38.22 |
| | w/o BP | 87.50 | 60.00 | 39.13 |
| | RetriKT-Retrieval | 94.64 | 66.00 | 47.87 |
| | w/o RL | 89.29 | 62.00 | 41.57 |
| | w/o R$_{accuracy}$ | 91.07 | 62.00 | 45.32 |
| | w/o R$_{diversity}$ | 87.50 | 61.00 | 43.24 |
| | w/o BP | 89.29 | 62.00 | 44.98 |
| BERT$_4$ | RetriKT-KD | 92.86 | 72.00 | 58.16 |
| | w/o RL | 89.29 | 67.00 | 53.18 |
| | w/o R$_{accuracy}$ | 91.07 | 65.00 | 57.78 |
| | w/o R$_{diversity}$ | 89.29 | 69.00 | 57.01 |
| | w/o BP | 91.07 | 66.00 | 57.27 |
| | RetriKT-Retrieval | 94.64 | 74.00 | 60.77 |
| | w/o RL | 92.86 | 69.00 | 56.45 |
| | w/o R$_{accuracy}$ | 92.86 | 69.00 | 59.94 |
| | w/o R$_{diversity}$ | 91.07 | 70.00 | 58.25 |
| | w/o BP | 92.86 | 69.00 | 57.82 |

Table 3: Ablation studies of different components utilizing BERT$_2$ and BERT$_4$ as the small-scale models on datasets CB, COPA and CoLA.

duct a series of ablation studies. We remove reinforcement training (w/o RL), accuracy reward R$_{accuracy}$, diversity reward R$_{diversity}$, and length penalty BP, and evaluate the results on CB, COPA, and CoLA, as shown in Table 3. Several key observations can be drawn from the experimental results. Firstly, it is evident that the model performance of both RetriKT-KD and RetriKT-Retrieval declines without RL or any component of RL. This highlights the effectiveness of our reward function design and RL, which enhance the generation quality of the LLM. Secondly, when any component of RL is removed, the model performance on CoLA shows relatively smaller decreases compared to CB and COPA. This suggests that the generation quality of the LLM has a more pronounced impact on smaller datasets, while larger datasets exhibit a certain level of robustness. However, RL consistently improves the generation quality of the LLM, thereby enhancing overall model performance. Finally, it is worth mentioning that RetriKT-Retrieval consistently outperforms RetriKT-KD across almost all settings, further validating the effectiveness of retrieval-based knowledge transfer for small-scale models.

| Metric | Model | WiC | CB | COPA | RTE | BoolQ | CoLA |
|--------|-------|-----|-----|------|-----|-------|------|
| Self-Bleu | RetriKT | 0.912 | 0.948 | 0.970 | 0.923 | 0.884 | 0.959 |
| | RetriKT w/o RL | 0.945 | 0.968 | 0.983 | 0.937 | 0.939 | 0.973 |
| Cross Entropy | RetriKT | 0.734 | 0.781 | 0.786 | 0.781 | 0.760 | 0.579 |
| | RetriKT w/o RL | 0.758 | 0.896 | 0.796 | 0.794 | 0.779 | 0.618 |

Table 4: Diversity and accuracy analysis for the generated knowledge. Self-Bleu reflects the diversity of knowledge, while cross entropy reflects the accuracy of knowledge. In both metrics, smaller values indicate better performance.

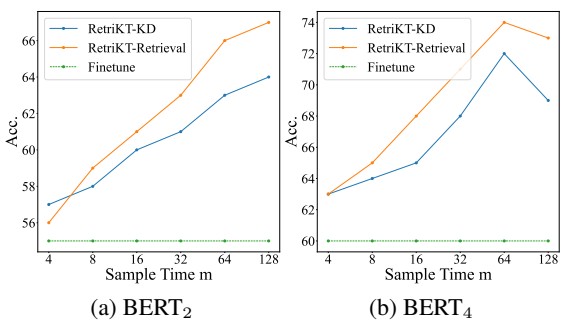

(a) BERT$_2$       (b) BERT$_4$

Figure 2: Effect of the sample times $m$. Results are the model performance on the COPA dataset using BERT$_2$ and BERT$_4$. The Finetune method is trained by the original dataset so it is independent of $m$.

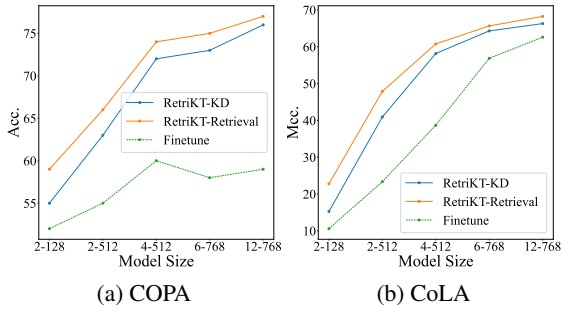

(a) COPA       (b) CoLA

Figure 3: Effect of the model size. x-axis is the number of layers and hidden size. Results are the model performance on the COPA and CoLA datasets.

## 5.3 Detailed Analysis

**Effect of the Sample Times** We conduct experiments on the COPA dataset to investigate the impact of the sample time $m$ on model performance. As shown in Figure 2, with a small value of $m$, the performance of both RetriKT-KD and RetriKT-Retrieval is comparable. However, as $m$ increases, we observe a significant improvement in the performance of RetriKT-Retrieval compared to RetriKT-KD. This finding suggests that small-scale models, constrained by their limited parameters, are unable to effectively memorize such a large amount of knowledge. Therefore, retrieval-based knowledge transfer proves to be a more favorable ap-

proach. Furthermore, as $m$ continues to increase, we observe a corresponding increase in the performance of both RetriKT-KD and RetriKT-Retrieval, reaching a certain threshold. This indicates that increasing the number of generated samples can enhance the diversity of knowledge. However, due to the limited number of keywords extracted from the original dataset, the diversity of knowledge plateaus once the generation threshold is reached. Consequently, the model performance does not show further improvement beyond this point.

**Effect of the Model Size** We conduct an investigation into the impact of model size on performance using two datasets, COPA and CoLA. Our experimental results shown in Figure 3 reveal that when employing a smaller model, RetriKT-Retrieval consistently outperforms RetriKT-KD, showcasing the superiority of retrieval-based knowledge transfer for small-scale models. Additionally, we observe that both RetriKT-Retrieval and RetriKT-KD consistently outperform the training of the small-scale model solely fine-tuning by the original dataset. This observation further emphasizes the effectiveness of knowledge extraction from the LLM and its application in enhancing model performance.

**Accuracy and Diversity** In this section, we examine whether the LLM trained through reinforcement learning can generate more accurate and diverse knowledge. To evaluate the diversity of knowledge, we employ the Self-Bleu metric on the generated samples, while the accuracy of knowledge is measured using the cross-entropy between the probability distribution predicted by the reward model and the label generated by the LLM. In both metrics, smaller values indicate better performance. As shown in Table 4, the results demonstrate that training the LLM through reinforcement learning leads to the generation of more accurate and diverse knowledge across all datasets. This outcome highlights the effectiveness of our reward function de-

sign. Additionally, we observe that larger datasets tend to yield samples with smaller Self-Bleu metrics. We conjecture that this phenomenon is a result of the larger dataset's ability to extract a wider range of diverse keywords, thereby enabling the generation of more varied knowledge.

## 6 Conclusion

Our study tackles the task of compressing LLMs and introduces a pioneering compression paradigm called Retrieval-based Knowledge Transfer. This approach efficiently transfers the knowledge of LLMs to small-scale models by creating a knowledge store, enabling the small model to retrieve pertinent information during inference. Through extensive experiments conducted on commonly used benchmarks, we demonstrate that our framework substantially improves the performance of small-scale models by leveraging the knowledge contained within LLMs. In future research, we plan to investigate the application and performance of our proposed method on even larger language models, such as T5-11B.

## Limitations

In this section, we discuss the limitations of our work as follows. Firstly, due to limited computational resources, we did not attempt experiments with larger models such as T5-11B. Furthermore, resource constraints constrained our grid search for hyperparameters within a limited range, which potentially leaves room for enhancing the metrics showcased in this paper. Secondly, our proposed method requires the establishment of a knowledge store. Compared to other model compression methods, our approach introduces additional storage space and incurs slightly additional time for retrieval

## Acknowledgements

This work is supported by National Key R&D Program of China (No. 2021YFC3340303) and National Natural Science Foundation of China (NSFC Grant No. 62122089). Jingang Wang is funded by Beijing Nova Program (Grant NO. 20220484098). We sincerely thank all reviewers for their valuable comments and suggestions, which are crucial for improving our work. We would also like to acknowledge Angela Li for her contributions in creating the figures used in this work.

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

## A  Training Details for PPO

In this section, we present the training details of PPO in Table. The meaning of these hyperparameters is detailed in trl library (von Werra et al., 2020) and NLPO (Ramamurthy et al., 2022).

## B  Preprocessed Examples

In this section, we provide examples of our preprocessing for each of the data sets we consider.

### B.1  boolq

**Original input:**

**Question:** `Can you have a skunk as a pet in canada`

**Passage:** `Skunks as pets – Canadian pet skunks must be purchased from a USDA-certified breeder in the United States. An import permit is required from the Canadian Food Inspection Agency to bring the skunk into the country. The skunk must be spayed or neutered, and receive a microchip implant or tattoo. A vet check fee must also be paid. It is illegal to keep striped skunks as pets in Canada.`

**Processed input:** `question: Can you have a skunk as a pet in canada passage: Skunks as pets – Canadian pet skunks must be purchased from a USDA-certified breeder in the United States. An import permit is required from the Canadian Food Inspection Agency to bring the skunk into the country. The skunk must be spayed or neutered, and receive a microchip implant or tattoo. A vet check fee must also be paid. It is illegal to keep striped skunks as pets in Canada.`

**Label:** `{False: 0; True: 1}`

### B.2  CB

**Original input:**

**Hypothesis:** `Valence was helping`

**Premise:** `Valence the void-brain, Valence the virtuous valet.  Why couldn't the figger choose his own portion of titanic anatomy to shaft? Did he think he was helping?`

**Processed input:** `hypothesis: Valence was helping premise: Valence the void-brain, Valence the virtuous valet. Why couldn't the figger choose his own portion of titanic anatomy to shaft? Did he think he was helping?`

**Label:** `{entailment: 0; contradiction: 1; neutral: 2}`

### B.3  CoLA

**Original input:**

**Sentence:** `John made Bill master of himself.`

**Processed input:** `sentence: John made Bill master of himself.`

**Label:** `{unacceptable: 0; acceptable: 1}`

### B.4  COPA

**Original input:**

**Question:** `effect`
**Premise:** `Political violence broke out in the nation.`
**Choice 1:** `Many citizens relocated to the capitol.`
**Choice 2:** `Many citizens took refuge in other territories.`

**Processed input:** `premise:    Political violence broke out in the nation. choice1: Many citizens relocated to the capitol.  choice2: Many citizens took refuge in other territories. question: effect`

**Label:** `{choice1: 0; choice2: 1}`

### B.5  RTE

**Original input:**

**Sentence 1:** `A smaller proportion of Yugoslavia's Italians were settled in Slovenia (at the 1991 national census, some 3000 inhabitants of Slovenia declared themselves as ethnic Italians).`
**Sentence 2:** `Slovenia has 3,000 inhabitants.`

| Hyperparameters | WiC | CB | COPA | RTE | BoolQ | CoLA |
|---|---|---|---|---|---|---|
| learning rate | 2e-3 | 2e-3 | 2e-3 | 2e-3 | 2e-3 | 2e-3 |
| batch size | 128 | 64 | 64 | 128 | 128 | 128 |
| min batch size | 32 | 16 | 16 | 32 | 32 | 32 |
| epoch | 20 | 100 | 100 | 50 | 10 | 20 |
| ppo epoch | 4 | 4 | 4 | 4 | 4 | 4 |
| sample number | 4 | 4 | 4 | 4 | 4 | 4 |
| initial kl coeff | 0.001 | 0.001 | 0.001 | 0.001 | 0.001 | 0.001 |
| target kl | 6 | 6 | 6 | 6 | 6 | 6 |
| value function coeff | 0.5 | 0.5 | 0.5 | 0.5 | 0.5 | 0.5 |
| clip ratio | 0.2 | 0.2 | 0.2 | 0.2 | 0.2 | 0.2 |
| discount factor | 0.99 | 0.99 | 0.99 | 0.99 | 0.99 | 0.99 |
| gae lambda | 0.95 | 0.95 | 0.95 | 0.95 | 0.95 | 0.95 |

Table 5: Hyperparameters for PPO

**Processed input:** `sentence1: A smaller proportion of Yugoslavia's Italians were settled in Slovenia (at the 1991 national census, some 3000 inhabitants of Slovenia declared themselves as ethnic Italians). sentence2: Slovenia has 3,000 inhabitants.`

**Label:** `{entailment: 0; not_entailment: 1}`

### B.6 WiC

**Original input:**

**POS:** `N`
**Sentence 1:** `It was the deliberation of his act that was insulting .`
**Sentence 2:** `The deliberations of the jury .`
**Word:** `deliberation`

**Processed input:** `sentence1: It was the *deliberation* of his act that was insulting. sentence2: The *deliberations* of the jury. word: deliberation`

**Label:** `{False: 0; True: 1}`

## C Faithfulness of the Generated Knowledge

To further validate the faithfulness of the generated knowledge, we gathered five volunteers to assess the faithfulness of the generated texts for datasets WiC, COPA, and CoLA. Each text will be given a rating from 1 to 5. We randomly sampled 100 texts from each dataset and calculate the average as the

| WiC | COPA | CoLA |
|---|---|---|
| 4.61 | 4.36 | 4.49 |

Table 6: The faithfulness score of the generated texts for datasets WiC, COPA and CoLA.

final faithfulness score. The results of the faithfulness evaluation are shown in Table 6, showing that the faithfulness of generated texts for each dataset is quite high.