# OpenReview forum: "Retrieval-based Knowledge Transfer: An Effective Approach for Extreme Large Language Model Compression"
_EMNLP/2023/Conference — EMNLP 2023 Findings_

### Official Review · Reviewer_6ByT · 2023-07-31

**Typos Grammar Style And Presentation Improvements:** None as far as I know
**Soundness:** 4

**Excitement:**

4: Strong: This paper deepens the understanding of some phenomenon or lowers the barriers to an existing research direction.

**Missing References:**

Knowledge Distillation of Large Language Models. Yuxian GU, Li Dong, Furu Wei, Minlie Huang. Arxiv 2023.

**Paper Topic And Main Contributions:**

This paper presents an interesting work on model compression. The authors propose to transfer knowledge from larger language models to smaller ones with a multi-step approach: tuning in-domain soft prompts, enhancing model stability with reinforcement learning, creating knowledge stores, and training the small-sized models. The experiments are conducted on various GLUE and SuperGLUE datasets with T5-3B as the teacher and show good performance improvement on very small-sized BERT-7.2M and BERT-13.5M.

**Questions For The Authors:**

Please refer to Reasons to Reject, thank you!

**Reasons To Accept:**

1. This paper presents an interesting work. The idea to disentangle and then combine the knowledge base building and embedding learning is well-motivated.

2. The writing is very clear and easy to follow, with good figures and tables presented.

3. The proposed method is evaluated by a diverse set of experiments (GLUE, SuperGLUE, Generation, and Ablation).

**Reasons To Reject:**

1. the teacher model used in the paper is a T5-3B model, which is not an “extreme large” language model as said in the title. This is an overclaim.

2. The student model is only tested with very small BERT models (around 1/10 of BERT-base). It raises the concern of the real-world applicability or extendibility of the proposed method. In (Gu et. al., 2023), 100M-700M parameter models are selected as students for a GPT-2-1.5B teacher. Their experiments also include 1.5-6B student models distilled from OPT-13B, which can be claimed as a large model.
On the other hand, what about T5 models that belongs to the family of the teacher model? It seems that there is no justification for why small-sized BERT-alike models are selected as the student.

3. Since the teacher model is not large, it is questionable that how well the generated knowledge base is in terms of faithfulness. Some auxiliary evaluation seems necessary to validate this step of the idea.

4. Since the retrievability is claimed to be improved, it seems natural to extend the experiments from NLU (e.g., COPA, CoLA, etc.) to other sorts of datasets requiring much retrievability, such as open-domain question answering (e.g., web questions, natural questions, etc.).

**Reproducibility:**

3: Could reproduce the results with some difficulty. The settings of parameters are underspecified or subjectively determined; the training/evaluation data are not widely available.

**Reviewer Confidence:**

3: Pretty sure, but there's a chance I missed something. Although I have a good feel for this area in general, I did not carefully check the paper's details, e.g., the math, experimental design, or novelty.

---

> ### Author Rebuttal · Authors · 2023-08-28
>
> We sincerely thank you for your recognition of our work and the valuable feedback. We will provide a detailed explanation based on your concerns.
>
> #### **Q1: Regarding the T5-3B as the teacher model.**
>
> **A1:** Thank you for pointing it out. We would like to provide some clarification here. Firstly, due to constraints in computational resources, our experiments were exclusively conducted on T5-3B, as indicated in the 'Limitations' section of our paper. Secondly, when we refer to 'extremely large,' we are actually alluding to the significant compression applied to the teacher model. In essence, it means an exceptionally high compression ratio. For instance, transitioning from 3B to 7.2M parameters represents an extreme compression, with a compression rate of 0.24\%. We will provide additional elaboration on this point in our revision.
>
> #### **Q2: Regarding very small BERT models as the student models.**
>
> **A2:** Thank you for the insightful question. We'd like to provide more explanations as follows.
>
> - We follow established methods for model compression, such as AD-KD (Liu et al.) and MSGKD (Wu et al.), which employ the BERT model as the student model.
> - Our proposed method is primarily designed for scenarios requiring significant compression ratios. Therefore, we conducted experiments using very small student models. These experimental outcomes serve to demonstrate the effectiveness of our approach in high compression scenarios.
> - Given that BERT offers pre-trained models of varying sizes (as released by Turc et al.), we selected BERT as the student model instead of T5.
>
> We appreciate your suggestions, and will supplement experiments on more student models.
>
> #### **Q3: Regarding the faithfulness of the generated knowledge.**
>
> **A3:** Thank you for the great suggestion. In Table 4, we demonstrate that our generated knowledge becomes more accurate and diverse after PPO training. To further validate the faithfulness of the generated knowledge, we gathered five volunteers to assess the faithfulness of the generated texts for datasets WiC, COPA, and CoLA. Each text will be given a rating from 1 to 5. We randomly sampled 100 texts from each dataset and calculate the average as the final faithfulness score. The results of the faithfulness evaluation are as follows, showing that the faithfulness of generated texts for each dataset is quite high.
>
> | WiC | COPA | CoLA |
> | :----:  | :----: | :----:   |
> | 4.61 | 4.36   | 4.49    |
>
>
> #### **Q4: Regarding extending the experiments from NLU to other sorts of datasets requiring much retrievability.**
>
> **A4:** In this work, we conduct experiments on NLU tasks to get a direct comparison with previous SOTA works. We appreciate your great suggestion to extend the experiments to other types of datasets requiring much retrievability. We plan to evaluate our approach on open-domain QA datasets.
>
> Thanks again for your constructive feedback. Please let us know if you have any further questions, and we are happy to discuss further.

---

### Official Review · Reviewer_CJV8 · 2023-08-04

**Soundness:** 3

**Excitement:**

3: Ambivalent: It has merits (e.g., it reports state-of-the-art results, the idea is nice), but there are key weaknesses (e.g., it describes incremental work), and it can significantly benefit from another round of revision. However, I won't object to accepting it if my co-reviewers champion it.

**Paper Topic And Main Contributions:**

This paper presents a method to enhance the performance of compressed models using a framework. The framework comprises soft-prompt-tuning an LLM that generates samples. These samples are refined through a reinforcement learning algorithm (PPO) using the soft-prompts model as an agent, and a reward function considering the diversity of samples and the accuracy of an auxiliary classifier. The latest is used to train the smaller model to generate closer embeddings, but also to generate the labels and store them together with embeddings in a database. During inference, the smaller model generates an embedding of the test sample, then queries the database. The top closest samples are then selected, and the answer is computed with a formula combining similarity scores along with the confidence of the classifier with respect to each label (like how softmax works). The label with the largest logit is then selected. This method seems to achieve good performance on some superglue/GLUE tasks, considering the size of the final model. However, it looks like it is computationally heavy and not very practical.

**Questions For The Authors:**

1.	I don’t quite understand the paragraph from line 48 to line 50. You mention knowledge distillation as a compression method, then claim that such “compression methods are not suitable for scenarios requiring high compression ratios, such as knowledge distillation…”. Besides, KD does not necessarily require high compression ratio, you can in theory use any ratio you want.
2.	I don’t quite understand the paragraph from line 259 to line 262: “unable to optimize these learning objectives due to lack of per-token differentiability”.
3.	How do we assess the impact of length penalty on the training? Penalizing length only does not mean that the model will generate sensical sentences.
4.	Paragraph 303->305: What s the LM you are talking about? The classifier or the large frozen model? Is there any equation representing the global loss function (including everything)?
5.	How is the whole training performed? Jointly? Sequentially?
6.	Line 407: are you using T5 as a teacher and Bert as student? These are two different architectures. Maybe I am not aware of it, but just out of curiosity, has this been done before in the literature?
7.	Line 446: you mean eq. 4?
8.	If I understand well, the small scale is only used to generate a representation that is close to those previously calculated by the reward model.  The decision only relies on the labels previously generated by the RM. Say the test sample and the retrieved sentences are by no means related to each other. That means the final answer is likely going to be based on answers to different questions. Correct?

**Reasons To Accept:**

1.	Authors present a novel method to train small-scale models on downstream tasks.
2.	Though the method is a combination of existing other methods, I haven’t seen it before in the literature.
3.	The reward function is well designed.
4.	Method seems to outperform some KD methods.

**Reasons To Reject:**

1.	Paper’s aspects hard to understand and writing needs to be improved. See questions for authors #1.
2.	Some aspects of authors idea are not clearly explained:
	a.	How is the generator trained? Pretrained only? Or finetuned as well?
	b.	PPO’s agent is not clearly described. It is quickly mentioned in lines 435-436.
	c.	How do you initialize small models?
3.	Method looks more over-engineered than theoretically sound; besides it is heavy and requires so many steps for a mitigated result:
	a.	Pre-training LLM
	b.	Finetuning the reward model.
	c.	Training the ensemble within the framework imagined by authors.
	d.	Generate a database (knowledge store).
	e.	Use data retrieval to enhance the smaller model performance.
The above steps should be conducted for each downstream task, which questions the usefulness of this method in practice.
4.	Missing some comparison baselines, such as simple finetuning. How does finetuning a randomly initialized/truncated BERT model perform compared to your method?
5.	RL section seems a bit handwavy: some maths/equations/details on the PPO agent would be more than welcome!
6.	Paper’s architecture is nearly the same as for PromDA’s, except for the processing of the samples, where the former uses PPO algorithm, while the latter uses consistency filtering.
7.	Method limited to text classification (for tasks) and models with sentence representations (like BERT with CLS token). This is not mentioned in the limitations.

**Reproducibility:**

4: Could mostly reproduce the results, but there may be some variation because of sample variance or minor variations in their interpretation of the protocol or method.

**Reviewer Confidence:**

3: Pretty sure, but there's a chance I missed something. Although I have a good feel for this area in general, I did not carefully check the paper's details, e.g., the math, experimental design, or novelty.

---

> ### Author Rebuttal · Authors · 2023-08-28
>
> We sincerely thank you for your careful review of our paper and the detailed feedback, which are crucial for improving our work. We discuss your raised points as follows:
>
> #### **Q1: Regarding how the generator is trained, PPO’s agent and the initialization of small models.**
>
> **A1:** Thank you for the questions. We would like to clarify these questions here.
>
> - For the training of the generator, we initialize it with the T5-3B model and augment it with randomly initialized soft prompts, as detailed in section 4.3. During the entire training process, we exclusively update the soft prompts while maintaining the fixed parameters of the T5-3B model. Our training procedure unfolds in two main phases. Initially, we employ supervised learning using appropriate datasets to train the generator, as elaborated in section 3.1. Subsequently, we fine-tune the generator using the reinforcement learning algorithm PPO, as outlined in section 3.2.
> - The PPO's agent is our generator. We train the generator by the reinforcement learning algorithm PPO, enabling it to generate more accurate and diverse knowledge.
> - Our small models are initialized from the BERT models released by Turc et al. (as mentioned in lines 429-433).
>
> We hope these additional information answers your questions. We will supplement these details in our revision.
>
> #### **Q2: Regarding the complexity and engineering of our methods.**
>
> **A2:** Thank you for the insightful question. Our approach indeed involves multiple steps during the training phase to facilitate effective knowledge transfer. However, once our model is trained, the inference stage becomes relatively straightforward, involving only sentence encoding followed by retrieval. Particularly for practical applications that pose challenges for smaller models, we consider our approach to be highly valuable and necessary.
>
> #### **Q3: Regarding the comparison with baselines, such as simple finetuning.**
>
> **A3:** Thanks for the suggestion. In fact, we have compared our approach with simple finetuning baselines in Table 2. The rows corresponding to BERT$_2$ (Student) and BERT$_4$ (Student) are the results of simple finetuning. We’ll make this more clear during revision.
>
> #### **Q4: Regarding more details of PPO algorithm.**
>
> **A4:** Since we directly apply the PPO algorithm without any refinement, we do not include its technical details in our paper. Thank you for the great suggestion. We will supplement the technical details of the PPO algorithm in the Appendix.
>
> #### **Q5: Regarding the architecture difference with PromDA.**
>
> **A5:** Thanks for the question. We would like to provide some clarification here. Our approach exhibits significant distinctions from PromDA in several key aspects:
>
> - Motivation: Our approach explores a novel compression paradigm, focusing on efficiently transferring knowledge from Large Language Models (LLMs) to smaller models. In contrast, PromDA's primary aim is to enhance few-shot tasks through data augmentation.
> - Model Architecture: We introduce the PPO algorithm to enhance the accuracy and diversity of generated knowledge. Additionally, we facilitate knowledge transfer to the smaller model through retrieval. The main commonality with PromDA is the utilization of the p-tuning v2 method (Liu et al.) for fine-tuning the generator.
>
> We hope this clarifies the distinctions between our approach and PromDA.
>
> #### **Q6: Regarding limitation to text classification and models with sentence representations.**
>
> **A6:** Thanks for pointing it out. Currently, our method is only applicable to text classification tasks, just like the baseline methods we compared with. In the future, we will expand our method to more types of tasks (such as token classification and generation). We will provide these discussions in the Limitations section as suggested.
>
> #### **Q7: Regarding the claim that "compression methods are not suitable for scenarios requiring high compression ratios, such as knowledge distillation**
>
> **A7:** Sorry for the confusion. We understand and agree that KD does not necessarily require a high compression ratio. The intention behind this paragraph is to emphasize that prevailing model compression techniques, including knowledge distillation, pruning, and quantization, might not be apt for situations requiring substantial compression ratios. Consequently, our approach is introduced as a solution to address this specific challenge.
>
> #### **Q8: Regrading unable to optimize these learning objectives due to lack of per-token differentiability**
>
> **A8:** The evaluation metrics of accuracy and diversity operate at the sentence level, which means they lack per-token differentiability. In this context, we'd like to elucidate why reinforcement learning is employed instead of supervised learning. We will provide additional clarification on this point in our revised work.
>
> #### **Q9: Regarding assessing the impact of length penalty on the training.**
>
> **A9:** Thank you for your question. As mentioned in section 3.2, we employ the length penalty as a mechanism to discourage the generation of excessively simple patterns by the generator. The ablation study results presented in Table 3 demonstrate that the inclusion of a length penalty indeed enhances the model's performance. It's important to note that while the length penalty doesn't guarantee that the model generates sensical sentences, its absence may lead to the generation of overly simplistic patterns. This underscores the importance and necessity of incorporating the length penalty into our approach.
>
> #### **Q10: Regarding the meaning of "LM" in the paragraph from line 303 to line 305.**
>
> **A10:** Sorry for the inconvenience. The LM loss represents the learning object in section 3.1 to train the Language Model (the generator). We will make this more clear during vision. We’ll also supplement a global loss function as suggested.
>
> #### **Q11: Regarding the whole training process.**
>
> **A11:** We train our models sequentially. **First**, we train the generator by supervised learning so that the generator can generate in-domain texts. **Subsequently**, we employ the reinforcement learning algorithm PPO to further tune the generator to make the generated texts more accurate and diverse. **Afterward**, a knowledge store will be constructed using the generator, and **finally**, the small model will be trained.
>
> #### **Q12: Regarding using T5 as a teacher and BERT as student models.**
>
> **A12:** Previous studies, including MINIDISC (Zhang et al.) and AutoDisc (Zhang et al.), have also employed a configuration where T5 serves as the teacher model and BERT as the student model. This configuration is quite common in the field. In our work, we adopt T5 as the teacher model for several reasons. T5 not only functions as the teacher for knowledge distillation, enabling comparisons with baseline methods, but also serves as a generative model for knowledge generation. The choice of BERT as the student model, as opposed to T5, is motivated by the availability of pre-trained BERT models of various sizes, readily accessible from the work of Turc et al. This allows for easy initialization of the student model with the desired architecture.
>
> #### **Q13: Regarding Line 446**
>
> **A13:** Yes, we meant Eq. 4 (instead of Eq. 3). Thank you for pointing it out, and we will revise accordingly.
>
> #### **Q14: Regarding the decision only relying on the labels previously predicted by the RM.**
>
> **A14:** Your understanding is indeed accurate. However, we’d like to reiterate on three key points:
> - **Diversity of the Knowledge Store:** We ensure diversity within the knowledge store through reinforcement learning and sampled generation. This approach helps ensure that the knowledge store covers a wide range of information, making it, at the very least, relevant, if not entirely specific, to the test examples.
> - **Limited Generalization of the Small Model:** Small models often exhibit limited generalization ability, especially when tasked with inferring samples that extend beyond the training set. This challenge is particularly pronounced for complex tasks.
> - **Enhancing the Small Model through Retrieval:** Enhancing the small model through retrieval serves to mitigate the training difficulty inherent to smaller models. This approach empowers the small model to effectively leverage the knowledge present within the Large Language Model (LLM). Consequently, retrieval-based knowledge transfer consistently outperforms direct knowledge distillation."
>
> These three points collectively underscore the advantages of our approach over direct knowledge distillation.
>
> Thank you again for your detailed review and constructive feedback. Please let us know if you have any further questions, and we are happy to discuss further.

---

### Official Review · Reviewer_MYsE · 2023-08-14

**Soundness:** 4

**Excitement:**

4: Strong: This paper deepens the understanding of some phenomenon or lowers the barriers to an existing research direction.

**Paper Topic And Main Contributions:**

The paper introduce a framework called Retrieval Based Knowledge Transfer for compressing LLMs for low-resource end-tasks. The framework has 4 parts:

- it trains soft-prompts for individual tasks in a supervised setting, using the method from P-tuning v2 paper, and for diversity in text generation, combines it with the idea from PromDA which results in training soft-prompts for generation from two views of end-task examples (keywords from the input text and output labels of the example)

- then it adds the PPO algorithm for tuning soft-prompts for generation diversity and label accuracy. Generations are evaluated using a reward model trained on task-specific data, rewarded for diverse generations measured via self-bleu metric and penalized for generation of short examples

- the datastore is constructed from 5 types of examples:
  - from the original dataset
  - examples/labels generated by conditioning prompts on keywords from examples of the original dataset
  - examples/labels generated by conditioning prompts on labels of the original dataset
  - examples/labels generated from soft-prompts tuned on labels of original examples and labels of generated examples in step 2
  - examples/labels generated from soft-prompts tuned on keyword of original examples and keywords of generated examples in step 3
  - the datastore is created with keys from the sentence representations of generated examples `and values from the output of the reward model for those examples

- the small LM is now trained by generating similarity scores for pairs of retrieved examples from the datastore and aligning them with scores generated for the same set of examples by the reward model

**Questions For The Authors:**

- Missing ethics section in the paper
- "the massive size of these models poses huge challenges for their deployment in real-world applications." - because this line has been used to motivate the work, I am adding this comment here:

the ability to generate more examples for a low-resource end-task is useful, however the selection of low-resource tasks should also be done keeping in mind the usefulness of training examples from that task for other tasks (aka usefulness of the task for OOD generalization)

If examples from a low-resource task do not contribute to OOD generalization, having such an elaborate knowledge transfer setup for a couple of hundred dev/test set examples seems excessive. If an end-task has just a couple of hundred examples, it's also less likely that the end-user queries will be a good distribution match with the examples from that end-task and highly specialized small LMs on such tasks will be redundant.

**Reasons To Accept:**

- strong performance against recent distillation baselines from Liu et al. (MSG-KD) and Wu et al. (AD-KD)

- usefulness of retrieval-based learning objective against vanilla distillation on generated examples (this effect is more pronounced for smaller models) (Table 1 results: RetriKT-KD vs RetriKT-retrieval)

- ablations justify the inclusion of both the PPO setup and the retrieval component of the model

**Reasons To Reject:**

- the method seems to contain way too many intermediate steps and has a lot of working parts to setup the entire retrieval mechanism with a small LM (complexity of the method is a downside against ease of adoption)
- but I agree this is not a super strong reason in itself and should be used more as a motivation for future work

**Reproducibility:**

3: Could reproduce the results with some difficulty. The settings of parameters are underspecified or subjectively determined; the training/evaluation data are not widely available.

**Reviewer Confidence:**

4: Quite sure. I tried to check the important points carefully. It's unlikely, though conceivable, that I missed something that should affect my ratings.

---

> ### Author Rebuttal · Authors · 2023-08-28
>
> We sincerely thank you for your recognition of our work and the valuable feedback. We will provide a detailed explanation based on your concerns.
>
> #### **Q1: Regarding the complexity of the method.**
>
> **A1:** Thank you for the great question. It's true that our approach involves complex computations during the training phase to ensure effective knowledge transfer. However, once our model is adequately trained, the inference stage becomes quite straightforward, involving only sentence encoding and retrieval. Particularly for tasks that pose challenges for smaller models, like those found in the SuperGLUE benchmark, we consider our approach to be highly valuable. We fully concur with the reviewer's observation that simplifying the training process of our approach represents a key avenue for future research. We intend to expand on this aspect with more in-depth discussion in the revision.
>
> #### **Q2: Regarding missing ethics section in the paper.**
>
> **A2:** The ethics section is not mandatory, and we think there are no ethical issues in our paper. Please let us know if there are any ethics we should mention in this work, and we are happy to discuss them in the ethics section.
>
> #### **Q3: Regarding the usefulness of the task for OOD generalization.**
>
> **A3:** Thanks for the excellent comment. We completely agree with the reviewer's perspective that the choice of low-resource tasks should be approached with careful consideration, taking into account the potential utility of training examples from such tasks for other tasks. Our motivation hinges on the extensive pre-training of LLMs on massive datasets, which equips them with a wealth of knowledge. When combined with our PPO training methodology, this synergy leads to the generation of diverse text outputs from the LLM. Consequently, our approach has the potential to enhance out-of-distribution generalization, making it particularly effective for addressing low-resource tasks.
>
> Thanks again for your review! Please let us know if you have any further questions, and we are happy to discuss further.

---

### Meta-Review · Area_Chair_eqCZ · 2023-09-15

**Recommendation:** 4

**Metareview:**

This paper proposes a new model compression procedure that uses, at inference time, information prepared with a larger model. The proposed method also incorporates further components such as soft prompts and PPO, and its main application scenario lies in low-resource settings. The reviewers highlight the good empirical performance of the proposed method at high compression rates, and the useful ablations. They note that the work is novel in its way of combining existing methods. One of the central criticisms reviewers raise is the complexity of the training/compression procedure, which requires several steps to train soft prompts, PPO, and construction of the database. While this may motivate future work (on simplifying said procedure) it could also limit practicality. Reviewers also mention that the proposed procedure has been evaluated on only NLU/classification tasks, which might be a limiting factor. The information provided via author response on the remaining points of criticism, such as comparison to other baselines, and faithfulness of the database are useful additional datapoints that should be incorporated in the next iteration of the paper.

---

### Decision · Program_Chairs · 2023-10-07

**Decision:**

Accept-Findings

**Comment:**

This paper proposes a new model compression procedure that uses, at inference time, information prepared with a larger model. The proposed method also incorporates further components such as soft prompts and PPO, and its main application scenario lies in low-resource settings. The reviewers highlight the good empirical performance of the proposed method at high compression rates, and the useful ablations. They note that the work is novel in its way of combining existing methods. One of the central criticisms reviewers raise is the complexity of the training/compression procedure, which requires several steps to train soft prompts, PPO, and construction of the database. While this may motivate future work (on simplifying said procedure) it could also limit practicality. Reviewers also mention that the proposed procedure has been evaluated on only NLU/classification tasks, which might be a limiting factor. The information provided via author response on the remaining points of criticism, such as comparison to other baselines, and faithfulness of the database are useful additional datapoints that should be incorporated in the next iteration of the paper.